# COVID-19 Update: The Golden Time Window for Pharmacological Treatments and Low Dose Radiation Therapy

**Seyed Mohammad Javad Mortazavi** [1,†], **B. F. Bahaaddini Baigy Zarandi** [2,†], **Abdollah Jafarzadeh** [3], **S. Alireza Mortazavi** [4] **and Lembit Sihver** [5,6,7,*]

1    Medical Physics and Engineering Department, School of Medicine, Shiraz University of Medical Sciences, Shiraz 7134845794, Iran; mortazavismj@gmail.com
2    Department of Pharmacology, School of Medicine, Shiraz University of Medical Sciences, Shiraz 7134845794, Iran; zarandi@sums.ac.ir
3    Department of Immunology, School of Medicine, Kerman University of Medical Sciences, Kerman 7616914115, Iran; jafarzadeh@kmu.ac.ir
4    Private Clinic, Shiraz 7134845794, Iran; alireza.mortazavi.med@gmail.com
5    Department of Radiation Physics, Technische Universität Wien, Atominstitut, 1040 Vienna, Austria
6    Department of Physics, Chalmers University of Technology, 41296 Gothenburg, Sweden
7    Department of Radiation Dosimetry, Nuclear Physics Institute of the CAS, 25068 Prague, Czech Republic
*    Correspondence: lembit.sihver@tuwien.ac.at
†    These authors contributed equally to this work.

**Simple Summary:** Low-dose radiation therapy has been introduced as a novel approach for COVID-19 patients. The anti-inflammatory properties of low-dose radiation, in contrast with the pro-inflammatory properties at high doses, has a key role in the management of COVID-19 pneumonia. While more than 10 studies have shown the therapeutic advantages of this treatment with the minimal risk of toxicity, due to factors such as radiophobia, there is still a reluctance to further investigate low-dose radiation therapy as an effective remedy against COVID-19-associated pneumonia. In this paper, the golden time window for pharmacological treatments and low-dose radiation therapy are addressed.

**Abstract:** At the beginning of the COVID-19 emergence, many scientists believed that, thanks to the proofreading enzyme of SARS-CoV-2, the virus would not have many mutations. Our team introduced the concept of radiation at extremely low doses in an attempt to establish selected pressure-free treatment approaches for COVID-19. The capacity of low-dose radiation to modulate excessive inflammatory responses, optimize the immune system, prevent the occurrence of dangerous cytokine storm, regulate lymphocyte counts, and control bacterial co-infections as well as different modalities were proposed as a treatment program for patients with severe COVID-19-associated pneumonia. There is now substantial evidence which indicates that it would be unwise not to further investigate low-dose radiation therapy (LDRT) as an effective remedy against COVID-19-associated pneumonia.

**Keywords:** COVID-19; SARS-CoV-2; selective pressure; mutation; antivirals

## 1. Introduction

The pros and cons of pharmacological treatment of COVID-19 have been addressed by different researchers around the globe [1–3]. In this context, COVID-19-related cytokine storm can be treated with proper anti-inflammatory agents, such as inhibitors of Janus kinases (JAKs), inhibitors of sphingosine kinase-2 (SK2), blocking anti-interleukin-6 (IL-6), anti-IL-6R, anti-tumor necrosis factor-α (TNF-α), and anti-IL-1 monoclonal antibodies as well as using of the traditional immunosuppressive corticosteroids that reduce the severity of the harmful systemic inflammatory reactions [4,5]. As reported by Zuckerman et al., coping with the shocking public health challenge of the COVID-19 pandemic, the

use of many unproven medications were initiated including, but not limited to, antiviral agents such as lopinavir/ritonavir, favipiravir, as well as other drugs such as barcitinib, hydroxychloroquin and chloroquine, plasma from convalescent patients, and anti-cytokine treatments such as tocilizumab [6].

Scavone et al., in their paper recently published in the British Journal of Pharmacology, have reviewed the main pharmacological properties of the currently used drugs, including those repurposed for treatment of COVID-19 (e.g., lopinavir/ritonavir, remdesivir, favipiravir, and tocilizumab). The antiviral, immunomodulatory and anti-inflammatory properties of current pharmacological treatments for COVID-19 were discussed in the mentioned review [7]. Despite its strength, the paper authored by Scavone et al. has some gaps. The first gap is that the findings of a massive WHO-funded randomized study conducted at 405 hospitals in 30 countries (11,330 adult patients) were ignored. The results of this WHO solidarity, recently published in the New England Journal of Medicine [8], show that remdesivir, hydroxychloroquine, lopinavir, and interferon regimens either had little or no effect on hospitalized COVID-19 patients, as indicated by overall mortality, start of ventilation, and hospital stay duration. The second omission comes from missing the key role of non-robust antivirals of imposing selective pressure on SARS-CoV-2 which drives the virus to evolutionary and adaptive mutations. However, previous studies show that there is a golden time window for antivirals (administered within 48 to 72 h post infection) [9]. Thus, in this case, the chance of viral evolution is considerably reduced (Figure 1).

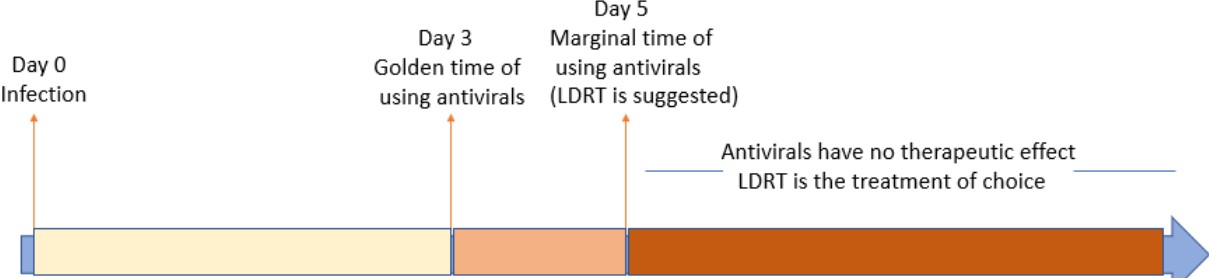

**Figure 1.** While early use of antivirals (when administered during the golden time window of 72 h post infection) may have a promising outcome and significantly decrease the risk of COVID-19, LDRT becomes a more promising treatment at later stages.

Now, the very large number of infected people (>500 million as of 14 April 2022) [10], the existence of immunocompromised patients (such as cancer patients and organ transplant recipients as well as patients on immunosuppressive therapy), along with treatment methods such as antivirals that exert selective pressure on SARS-CoV-2, increase the likelihood of viral evolution through adaptive mutations. As shown in Figure 2, this phenomenon increases the chance of the emergence of new variants of the virus. As shown in Figure 3, when an antiviral therapy is ineffective and hence unable to eradicate the virus, and in particular when more than millions of people are infected, the virus acquires a great opportunity to evolve through mutations. This problem may play a key role in limiting the success of infection control. The virus's properties must be taken into account when managing the SARS-CoV-2 pandemic. When about over 500 million people are infected with the virus, the widespread use of non-fully effective antiviral agents/vaccines can lead to unexpected outcomes.

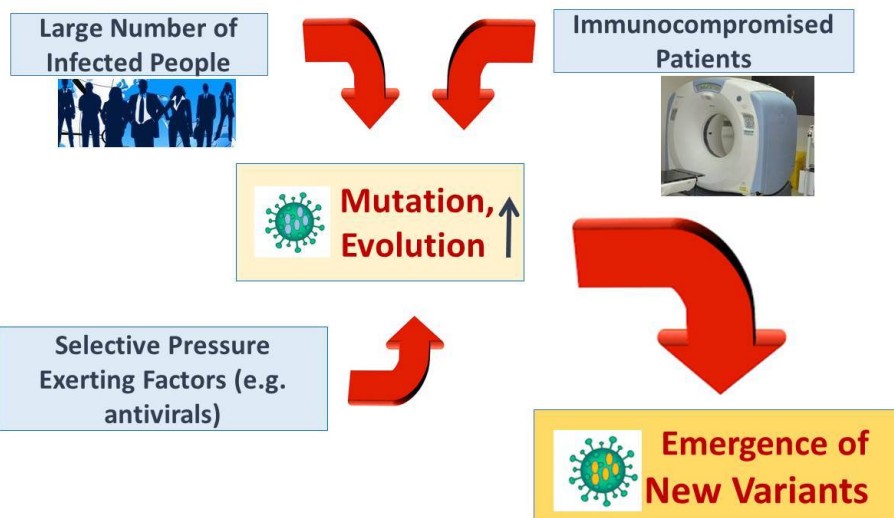

**Figure 2.** Factors such as the large number of infected people, the existence of immunocompromised patients (such as cancer patients), and treatment methods such as antivirals that exert selective pressure on SARS-CoV-2, increase the likelihood of viral evolution through adaptive mutations and hence increase the chance of emerging new variants.

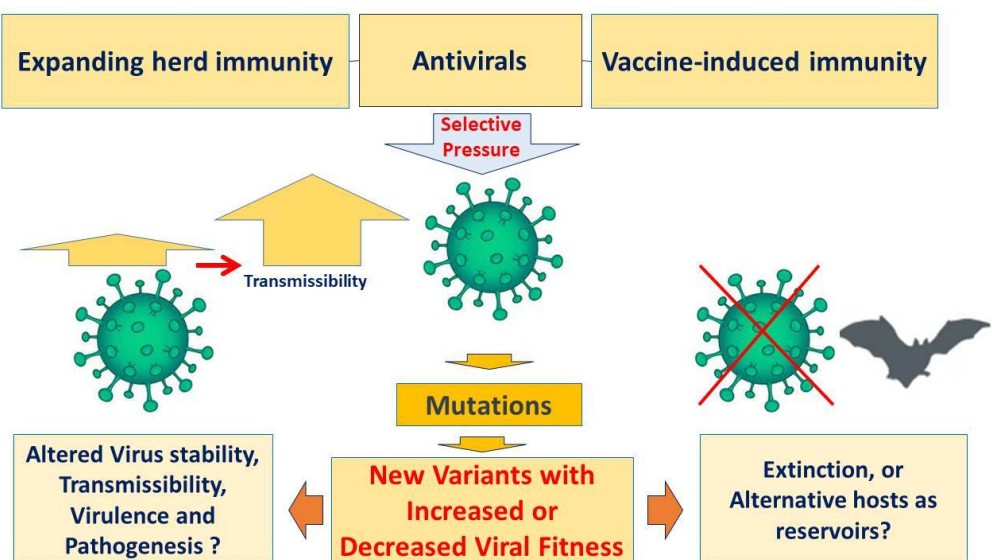

**Figure 3.** Non-robust antivirals can drive SARS-CoV-2 to evolution and the emergence of new variants by exerting selective pressure on the virus. Increased transmissibility, risk of jumping to other species, and resistance against antibodies and drugs can be associated with the emergence of new variants [11].

A recent report states: "Generally, coronaviruses are not sensitive to current antiviral drugs, and high concentrations of drugs effective on these viruses cannot be used in vivo" [12]. In line with our previous publications, Colson et al. have noted that, based on the reports of viral mutations in immunocompromised patients who received treatments such as remdesivir to treat prolonged COVID-19, it could be hypothesized that therapeutical approaches that fail to cure COVID-19 could favor the emergence of immune-escaping SARS-CoV-2 variants: "several cases have been reported for which the emergence of SARS-CoV-2 variants with mutations within the viral spike was evidenced in immunocompromised patients with prolonged SARS-CoV-2 infection who had received remdesivir and/or convalescent plasma or anti-spike antibodies [13]. The mutation rate was in some cases dramatically greater than that estimated for SARS-CoV-2 ($9.8 \times 10^{-4}$

substitutions/site/year, or 29.3 substitutions/genome/year)" [13]. Given this consideration, Colson et al. hypothesized that both remdesivir and convalescent plasma therapy (either alone or in combination) play a crucial role in the formation and selection of amino acid changes in the spike protein of SARS-CoV-2 [13]. Moreover, Torneri et al. note that "There are currently no potent and selective antivirals available against coronaviruses" [14]. Despite current debates over antiviral efficacy, Torneri et al. believe that the administration of effective antiviral drugs in combination with testing, isolation, and quarantine may significantly decrease the total number of patients and the peak incidence [14].

As mentioned above, our team has already discussed the significant concerns associated with the widespread use of non-robust antiviral drugs and the consequences of exerting strong selective pressures on the virus. At the beginning of the COVID-19 emergence, despite the fact that SARS-CoV-2 is an RNA virus, many scientists believed that thanks to the proofreading enzyme of SARS-CoV-2, the virus would not have many mutations. However, a wide number of SARS-CoV-2 mutations associated with high transmissibility were later observed. Since January 2020, WHO and its partners, have been monitoring and assessing the evolution of SARS-CoV-2. The emergence of new variants with increased risk to global public health during late 2020 prompted the characterization of specific Variants of Interest (VOIs) and Variants of Concern (VOCs) [15]. Presently in the US, two coronavirus subvariants known as BA.5 and BA.4 account for nearly 54% and 17% of the country's COVID cases (as of 7 July 2022), respectively. It is worth noting that the subvariants of BA.4 and BA.5 were originally detected in South Africa in January and February 2022, respectively. Mortazavi et al. have previously warned that due to key factors including, but not limited to, the vaccine inequity and the existence of millions of people living with HIV in Africa, a tragic catastrophe may occur in Sub-Saharan Africa that will possibly affect the entire world [16]. The subvariants of BA.4 and BA.5 have triggered the fifth wave of infections [17].

Our team introduced the concept of radiation at extremely low doses in an attempt to establish selected pressure-free treatment approaches for COVID-19 [18,19]. The capacity of low-dose radiation to modulate excessive inflammatory responses, optimize immune system, prevent the occurrence of dangerous cytokine storm, regulate lymphocyte counts, and control bacterial co-infections as well as different modalities were taken into account in our suggested treatment program for patients with severe COVID-19-associated pneumonia [20,21]. There is now substantial evidence which indicates that it would be unwise not to further investigate low-dose radiation therapy (LDRT) as an effective remedy against COVID-19-associated pneumonia [22].

## 2. Conclusions

While early use of antivirals (in particular, when administered during the golden time window of 72 h post infection) can show a promising outcome and significantly decrease the risk of COVID-19, LDRT becomes a more promising treatment at later stages. The majority of the trials conducted so far support the clinical advantages of LDRT for COVID-19. However, we suggest conducting very large-scale studies with advanced study design that fully removes or at least controls confounding variables such as concomitant administration of steroids or antiviral drugs.

**Author Contributions:** Conceptualization, S.M.J.M. and L.S.; literature review, S.M.J.M., B.F.B.B.Z., A.J., S.A.M. and L.S.; writing—original draft preparation, S.A.M.; writing—review and editing, S.M.J.M., B.F.B.B.Z., A.J., S.A.M. and L.S.; supervision, S.M.J.M. and L.S. All authors have read and agreed to the published version of the manuscript.

**Funding:** This research received no external funding.

**Institutional Review Board Statement:** Not applicable.

**Informed Consent Statement:** Not applicable.

**Data Availability Statement:** Not applicable.

**Acknowledgments:** The authors would like to thank the members of Scientists for Accurate Radiation Information (SARI) for their insightful comments.

**Conflicts of Interest:** The authors declare no conflict of interest.

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
