# Peer review of "COVID-19 Update: The Golden Time Window for Pharmacological Treatments and Low Dose Radiation Therapy"

_radiation, doi:10.3390/radiation2030020_

Round 1

Reviewer 1 Report

In the last 2 paragraphs (lines 84-90 and 91-104) adding some of quantitative values may make your statements more convincing. 

Author Response

Thank you very much for your valuable comments. We have added a simple summary, a conclusion, and we have added some more information in the last 2 paragraphs (lines 84-90 and 91-104 to may make our statements more convincing.  

Reviewer 2 Report

The manuscript is commentary but at the beginning I did not understand of what commentary it is. Now I think I understood the goal of the commentary, but in both situation: I really understood the goal or I am wrong, the goal of commentary should be stated clearly (and in the abstract as well). The commentary is for „Radiation” because authors published the article about Covid and LDRT in the Radiation in 2021. And the commentary is because since then several papers regarding pharmacological treatment of COVID-19 have been published. In these papers the considerations on the effectiveness of the pharmacological therapy has been shown as well as treatment influence on the mutation and evolution of the virus. The non-virus pressure LDRT is confronted with pharmacological treatment and authors shows LDRT advantages but I as a reader have impression the its virus mutation pressure-free character is the LDRT main advantage and this is not true. I would put accents slightly different showing that there is a lot of advantages of LDRT and one of them is its pressure-free quality.

The rest is interesting and perfect for me, no other remarks. I like the figures very much, as they are clear and colourful.

Author Response

Thank you very much for your valuable comments. We have added an abstract, a conclusion and a summary. We have tried to make is clearer that the non-virus pressure LDRT is confronted with pharmacological treatment and there is a lot of advantages of LDRT and one of them is its pressure-free quality.